# Evaluation and Functional Zoning of the Ecological Environment in Urban Space—A Case Study of Taizhou, China

**Haixia Zhao [1,\*], Xiaowei Jiang [2], Binjie Gu [1] and Kaiyong Wang [3]**

1   Nanjing Institute of Geography and Limnology, Chinese Academy of Sciences, Nanjing 210008, China; gubinjie21@mails.ucas.ac.cn
2   Jiangsu Environmental Engineering Technology Co., Ltd., Nanjing 210019, China; jiangxiaowei_89@163.com
3   Institute of Geographic Sciences and Natural Resources Research, Chinese Academy of Sciences, Beijing 100101, China; wangky@igsnrr.ac.cn
*   Correspondence: hxzhao@niglas.ac.cn; Tel.: +86-25-86882283

**Abstract:** Functional zoning provides a basis for establishing a regional development layout with clear functions, reasonable division of labor, and complementary advantages. In the process of urban development, a large number of behaviors such as occupying ecological land and generating a lot of pollution cause damage to the urban ecological environment. Functional zoning of the ecological environment has become an important tool used by the local and central governments to establish a harmonious relationship between socio-economic welfare and the ecological environment in recent years. Guided by the concepts and principles of ecological function zoning, this study applies and extends the methodological approach of ecological function zoning to the scale of urban space. Based on consideration of an evaluation of ecological environment sensitivity, ecosystem service function importance, and socio-economic coercion, this paper divides urban space into four types, namely: ecological environment restoration zone, ecological economy bearing zone, ecological environment preservation zone, and ecological environment protection zone, utilizing a mutually exclusive matrix classification. Taking Taizhou in Jiangsu Province as a pilot study, this paper verifies the actual application of a theoretical model and its technicalities, thus advancing the general case for function zoning of the ecological environment. Furthermore, it outlines measures for ecological environment protection and the industrial development orientation of each function area, thus providing a scientific basis for Taizhou's ecological city development and construction.

**Keywords:** ecological environment; functional zoning; GIS; matrix classification method; Taizhou

## 1. Introduction

Function zoning of the ecological environment is a key approach to identifying the characteristics, structure, and functions of a regional ecosystem, and it could reflect the main ecological function state and potential pressure on it [1]. The research on the Millennium Eco-environment Assessment Report issued by the United Nations shows that 60% of global ecosystem service functions have deteriorated, and meanwhile, it is estimated that the deterioration of ecosystem service functions will possibly be aggravated in the upcoming 50 years [2]. Establishing a technical system for environmental management based on the function zoning of the ecological environment is an inevitable development trend of ecological environment management, and also in the leading-edge direction of international environmental management. It is an efficient way to protect the ecological environment, which is an urgent task for rapid development areas such as those in China.

Related research started in the 1980s, and was been widely applied on a large regional scale [3], including some zoning work by the U.S. Environmental Protection Agency [4], Sierra Club [5], China Council for International Cooperation on Environment and Development [6], World Wildlife Fund International [7], and the Food and Agriculture Organization of the United Nations [8], forming a relatively perfect ecological function zoning index system and providing important references for follow-up research. In Europe, studies on functional zoning of ecological environments are gradually becoming quantified, with Janne Soininen quantifying ecological space through the analysis of socio-spatial structures [9] and Govaert Lynn using the importance of ecology and evolution to populations and communities as indicators of eco-evolutionary zoning [10]. In China, which is facing an increasingly austere ecological situation, Fu et al. [11,12] established principles, methods, and an index system for the comprehensive zoning of China's ecological environment; Yang and Li [13] made clear the basic partition of China's ecological regions and divided the whole country into 52 ecotypes, establishing a macroscopic framework for each region of the whole country to further develop ecological function zoning. In 2001, the State Environmental Protection Administration organized the preparation of the Interim Regulations on Ecological Function Zoning, which regulates the general principles, methods, procedures, contents, and requirements of provincial ecological function zoning, and is used to guide and normalize each province's development of ecological function zoning. Since then, research on ecological function zoning of various scales has been developing gradually. For example, provincial scale work includes Sichuan [14], Anhui [15], Fujian [8], and Xinjiang [16]; municipal scale work includes Qingdao [17] and Changchun [18]; cross-administrative regional scale work includes the Tai Lake Basin [19], the Three Gorges Reservoir Region [20], and Heilong River Basin [21], etc., and related research has enriched the methodology system of ecological function zoning at different scales. Meanwhile, scholars have constantly summarized and improved the zoning index system based on practice. For example, Yan and Yu [22], Liu et al. [23], and Liu and Chen [24] evaluated the index system of China's ecological function zoning and its application in related fields by taking ecological environment health, technical application, and environmental impact assessment on post-disaster reconstruction planning as orientation, respectively; in terms of the partition connotations, emphases and zoning methods, Sun et al. [25] made a comparison of Chinese and American ecological function zoning index systems, and found out that the role of human beings in an ecosystem and the ecosystem service function in China were highlighted, while the natural ecosystem was underlined in America. Early economic construction in China caused land use changes and uncontrolled expansion of urban areas, resulting in the conversion of a large amount of ecological land into land for construction. The land use change further damages the ecosystem function. Functional zoning of the ecological environment is conducive to promoting urban development in an orderly and scientific manner.

As a whole, many related research studies have been conducted. However, most are about the summarization of practical research, and less focus on innovation in ecological function zoning methods, and especially neglect the inclusion of scale difference of zoning methods. Present index systems for function zoning [26,27] of the ecological environment mostly lay particular emphasis on ecological sensitivity and the importance of ecosystem service function, but give less consideration to the stress effects of human activities on the ecological environment. However, along with the increase in the scope and strength of human activities' influences on the natural ecosystem, the ecosystem service function has been further damaged and weakened by human activities [28]. Referring to mature zoning methods at home and abroad, this paper particularly considers the stress effects of human activities on the ecological environment, constructs an innovative zoning evaluation index system, and improves the zoning method.

As a relatively under-developed city among the Yangtze River Delta urban agglomeration, Taizhou has a good ecological environmental background. However, it has worsened in recent years, along with the acceleration of urbanization, as polluting industries transferred from developed cities like Shanghai and South Jiangsu, etc. For example, in Suzhou, located in South Jiangsu, above-designated enterprises decreased from 13,538 in 2010 to 11,900 in 2020, as opposed to Taizhou where above-designated enterprises increased from 2531 in 2012 to 2861 in 2020. Therefore, it is suggested that fresh action should be carried out to solve the urban ecological environment problems of pollution, resources shortages, and ecological destruction. The purpose of this paper is to improve the basic theory of functional zoning by considering the stress of human activities on ecological environment for functional zoning based on the previous studies of ecological sensitivity and the importance of ecosystem service functions. Applying the method for ecological environment function zoning with Taizhou as a case study has a typical demonstration significance for medium-scale cities or regions.

## 2. Materials and Methods

### 2.1. Site Description

Taizhou is located in the central part of Jiangsu and is north of the Yangtze River Delta urban agglomeration, at 32°01′57″ N–33°10′59″ N and 119°38′24″ E–120°32′20″ E. Separated by the Yangtze River from Zhenjiang, Changzhou, Wuxi, and Suzhou in the south, close to developed Nanjing in the east, connected with Yangzhou in the west, and neighboring Yancheng in the north, Taizhou is at the junction of five channels flowing from the center of Jiangsu to the Yangtze River and sea, and is the conjunction area of the coastal and Yangtze River "T"—shaped industrial belt. The whole city consists of three districts (Hailing, Jiangyan, and Gaogang), has jurisdiction over three county-level cities (Xinghua, Jingjiang, and Taixing), and covers a total area of 5787.26 km$^2$, accounting for 5.64% of Jiangsu's total area. Except for Jingjiang having an independent massif, the whole city is in the alluvial plain of two water systems, namely the Yangtze River and Huai River (Figure 1). In the region are dense river networks interwoven horizontally and vertically, whereas in the north area, the topography is low-lying, and the water network is afferent-shaped, distributing many lakes. According to the data platform provided by the Taizhou government (http://tjj.taizhou.gov.cn/data/ accessed on 15 May 2022), since being established in 1996, Taizhou has achieved rapid economic development, and its regional gross product output increased from RMB 28.42 billion in 1996 to RMB 531.28 billion in 2020, representing an annual average growth rate of around 30.1%. In 2020, the total population reached 4,516,800, and the population density was 780 persons/km$^2$.

### 2.2. Function Zoning

Aiming to maintain the sustainability of habitat environment, conserve the safety of natural ecology, and optimize the pattern of ecological space, and through the regional element superposition analysis of factors such as ecological environment sensitivity, ecosystem service function importance, and socio-economic coercion, this study develops a mutually exclusive matrix classification for different spatial levels (Figure 2); it derives four types of function zones, namely ecological environment restoration zone, ecological economy bearing zone, ecological environment preservation zone, and ecological environment protection zone.

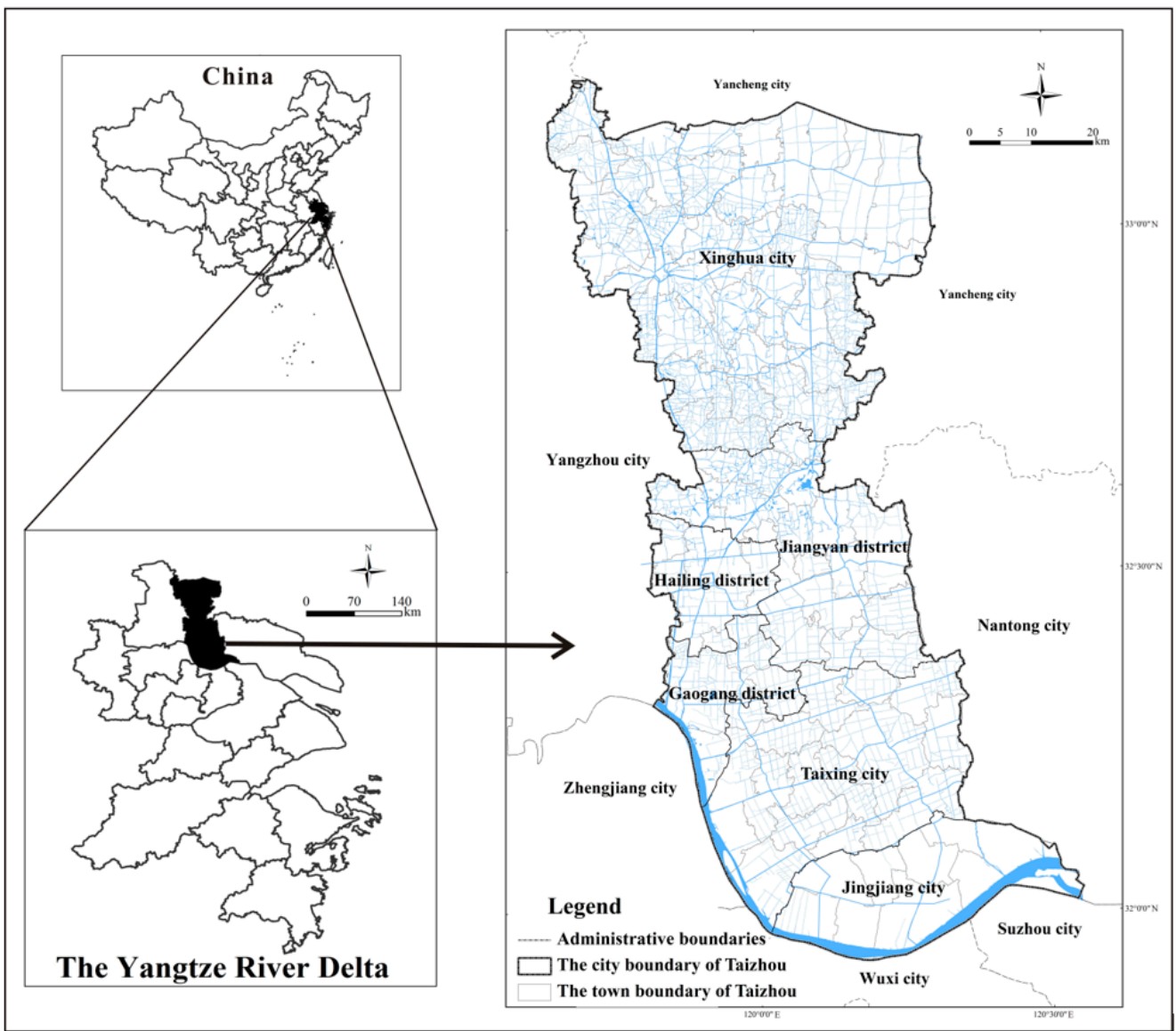

**Figure 1.** Location of the study area.

### 2.3. Indices and Weight

The selected indices not only reflect the characteristics of evaluation units' regional differences, but also have relative independence and non-replaceability. Wherein, the three primary indices, namely ecological environment sensitivity, ecosystem service function importance, and socio-economic coercion, jointly characterize the status of the regional ecological environment [29]. The ecological environment sensitivity index reflects the sensitive degree that the regional ecological environment factor is affected by the outside world, and stronger sensitivity of ecological environment will lead to weaker ability to resist external influences, stricter control on total pollution emission, and higher restriction on layout of polluting industries. Here, three secondary indices are mainly adopted for comprehensive analysis, namely the sensitivity of soil erosion, vulnerability to natural disasters, and sensitivity of water environment. The ecosystem service function importance index indicates the significance level of the natural environmental conditions and utilities provided by the ecosystem for human beings' survival [30], and the three secondary indices are mainly adopted for comprehensive characterization, namely the importance of water source conservation and drinking water source protection, the importance of natural and cultural heritage protection, and the importance of biodiversity and ecological environment

protection. The socio-economic coercion index takes into account the stress effects of human activities on the ecological environment, from factors such as pollution discharge, land use method, population gathering, industrial layout, and traffic development. The greater the degree of socio-economic coercion on the ecological environment, the more necessary it will be to control human activity intensity and adjust industrial layouts [31]. In terms of index selection, we shall mainly select indices of economic and social development for evaluation, and specifically, five secondary indices are selected: pollution discharge, construction land proportion, population density, industrial park area proportion, and road network density.

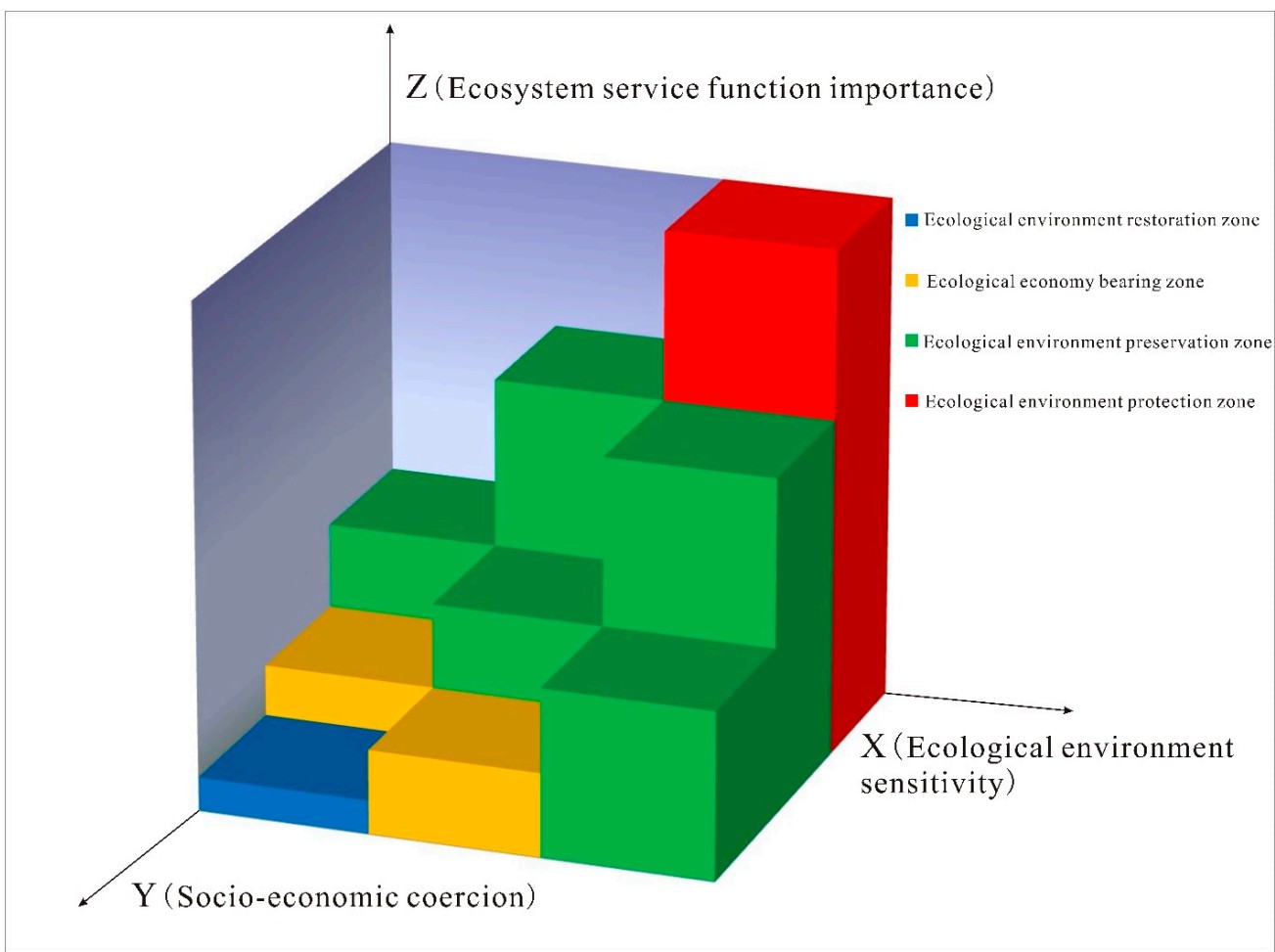

**Figure 2.** Matrix classification of regionalization.

The weighting was mainly determined by considering the difference and importance of the target factors. In order to set factors equitably and in a more scientific way, the scholar adopted analytic hierarchy process (AHP) and the Delphi method to determine the weight of each factor, as they would be more trusted if many experts evaluated them. AHP combines quantitative analysis with qualitative analysis, uses the decision maker's experience to judge the relative importance between the criteria that measure the achievability of each goal, and reasonably gives the weight of each criterion for each decision option, and uses the weights to find out the order of merit of each option [32]. Firstly, by means of AHP, 10 professors judged the relative importance of each index at each level, introduced appropriate scales, and quantified them with numerical values, in order to form a judgment matrix of each level. Secondly, combining factors such as the natural background, ecological environment, and current social and economic state of the Taizhou administration area, and by means of AHP, the method calculated the importance degree of each factor and took the result as initial weighting. Then, three experts in the field

of ecological and environmental protection were invited to rate the importance of their indicators, and 80 questionnaires were distributed to professional technical personnel in the ecological and environmental protection department to rate the importance of each factor, analyze the validity and astringency results of exports' scoring, ascertain the importance sequence of each factor, and prepare a weight judgment matrix table [33]. Finally, the weight of factors was calculated according to the judgment matrix (Table 1).

**Table 1.** Regionalization indices and their weights.

| Primary Indices | Secondary Indices | Weight | Index Factor |
|---|---|---|---|
| Ecological environment sensitivity | Sensitivity of soil erosion | 0.248 | Conduct comprehensive estimate by combining with factors such as regional rainfall, soil type, landform and topography, and vegetation coverage, etc. |
| | Vulnerability to natural disasters | 0.324 | Conduct comprehensive evaluation by selecting the factors like low-lying land, flood disaster, earthquake, ground setting, surface collapse, and ground fissure, etc. |
| | Sensitivity of water environment | 0.428 | Select drinking water head site, water environment function area's water quality objective, and land use type, etc., as evaluation factors. |
| Ecosystem service function importance | Importance of water source conservation and drinking water source protection | 0.320 | Select the factors like drinking water source reserve, important wet land, clean water channel maintenance area, and special ecological industry park, etc., to reflect ecological significance level. |
| | Importance of natural and cultural heritage protection | 0.345 | Select the factors like scenic spot, forest park, historical and cultural area, cultural relics and historical sites, etc., for evaluation. |
| | Importance of biodiversity and ecological environment protection | 0.335 | Adopt species quantity, plant coverage degree, and important protected specifies distribution, etc., for evaluation, and vegetation coverage index used as token of the purpose for reason of the influence of data. |
| Ecological environment stress | Pollution discharge pressure | 0.285 | Conduct comprehensive evaluation based on industrial point-sourced pollution, livestock and poultry breeding pollution, and urban domestic pollution. |
| | Urban expansion pressure | 0.215 | The proportion of the land use area for unit grid construction is used as a token. |
| | Population gathering pressure | 0.204 | The population density per grid is used as a token. |
| | Industrial layout pressure | 0.186 | The area proportion of planning zones for national, provincial, municipal, and township industrial park is used as a token. |
| | Traffic layout pressure | 0.110 | The current state of expressways, national and provincial highways, and the layout of planned road network are used as tokens. |

Based on the above, the study standardized the value of grid unit factors and calculated the weighted sum, and divided the ecological environment sensitivity index, ecosystem service function importance index, and socio-economic coercion index into high, relatively high, medium, relatively low, and low types, respectively, by means of a gradually hierarchical clustered merging and homoplasy analysis. Standardization and weighted sum formulas are as shown below:



$$X_{ij} = \frac{x_{ij} - x_{i\min}}{x_{imax} - x_{i\min}} \tag{1}$$

wherein $X_{ij}$ indicates the standardized numerical value of unit $j$ of index $I$; $x_{imax}$ indicates the maximum value of index $I$; $x_{imin}$ indicates the minimum value of index $I$, and $x_{ij}$ indicates the initial value of unit $j$ of item $i$.

$$A_i = \sum_{j=1}^{n} X_{ij} \times P_j \tag{2}$$

wherein $A_i$ indicates the ecological environment sensitivity index (or ecosystem service function importance index, socio-economic coercion index) of unit $i$; $X_{ij}$ indicates the value of factor $j$ of unit $i$, and $P_j$ indicates the weight of factor $j$.

*2.4. Evaluation Unit Division*

Evaluation units are mainly determined by considering the basic unit demand and factor characteristics analysis of zoning management. A smaller unit scale makes an analytic result more approximate to objective existence, but brings more difficulty in data acquisition and operation. This study takes $1 \times 1$ km grid as the basic spatial evaluation unit, and divides the study area into 6147 grid units. Natural factors are evaluated by taking natural boundaries as evaluation unit, while the data on economy and environmental pollution, and so on, are segmented into grid units based on township administration units. However, the conversion from polygon evaluation units to grid units is required for the evaluation of natural factors, and spatial superposition analysis is used for unifying evaluation units. The conversion formula is as shown below:

$$D_i = \sum_{j=1}^{n} (N_j \times A_{ij}) / A_i \tag{3}$$

wherein $D_i$ indicates the natural factors evaluation index of grid evaluation unit $i$; $A_{ij}$ indicates the area of level-$j$ natural factors in grid evaluation unit $i$; $A_i$ indicates the total area of grid unit $i$, and $N_j$ indicates the weight of level-$j$ natural factor.

*2.5. Data Source and Processing*

The data adopted include social economic statistics, natural factors, and environmental pollution discharge data. Whereas population and economic data is taken from the 2013 Taizhou City Statistical Yearbook, where the data are the official population and economic statistics for 2012 published by the Taizhou City Bureau of Statistics in 2013, the natural factor data, such as ecological importance, are mainly drawn from relief maps, remote sensing images, and current land use maps; the construction land data are from remote sensing interpretation and detailed surveys of land use change; the communication network is made according to current traffic state and planning charts; disaster data are from the national land department's geological disaster investigation and related research; and environmental pollution discharge data are from the environmental protection department's environmental statistics data. Among them, the remote sensing image is the data of a GF-2 satellite with a resolution of 2 m; the land use data are the current vector database of land use in Taizhou City in 2012 provided by Taizhou Bureau of Land and Resources, and the land use type classification is based on the land classification of the second national land survey, including: arable land, garden land, forest land, grassland, urban, village, and industrial and mining land, transportation land, water and water conservancy facilities land, and other land.

For the intrinsic diversification and broad sources of data, different data shall be processed according to different index characteristics, as shown below:

Spatial characteristics, such as natural disaster area, water source conservation area, water environment function area, built areas, and so on, are segmented to grid units through operation of intersection in ArcGIS 10.2 software, with area percentage as value; and as for the grids that cross different grades but the same type of factors, the value of corresponding factors is calculated with a weighted method.

Point-shaped pollution data, such as the industrial point-sourced pollution, are simulated by dint of the Kernel module for ArcGIS spatial analysis, and segmented to each grid through weighted sum analysis of different pollution densities, in order to represent the spatial distribution model of point-sourced pollutant discharge. Among them, the search radius in Kernel is selected according to the total number and main distribution of industrial point-sourced pollution in Taizhou, using the optimal radius in the Kernel module algorithm.

Social economy and surface-sourced pollution data for the township administration units, such as population and urban domestic pollution, need to be segmented to each evaluation grid, so that the spatial difference of different areas may be reflected by evaluating the indices such as the unit grid's population and pollution discharge.

Spatial expansion indices, such as traffic facilities, are calculated by establishing a buffer zone module with ArcGIS 10.2 software. Firstly, based on the linear data of main roads, we shall establish multi-level buffer expansion areas, and evaluate different levels of buffer zones; then segment the buffer zones into each grid unit by means of intersect operation, calculate the weighted sum of the area of different-level buffer zones in each grid unit, then take the result as a corresponding index of the grid unit. Among them, the road buffer zone is set up with a radius of 50 m, 100 m, and 200 m buffer zones.

## 3. Results

### 3.1. Ecological Environment Sensitivity Evaluation

#### 3.1.1. Soil Erosion Sensitivity

Soil erosion is affected and restricted by multiple factors. The purpose of soil erosion sensitivity evaluation is to measure the possibility of soil erosion occurrence considering natural factors. According to the measurement of Taizhou's soil erosion sensitivity with the RUSLE soil erosion model, the whole city's soil erosion sensitivity is generally light. The areas of relatively high sensitivity are mainly Taixing's north area, Jiangyan's south area, Hailing District, Gaogang District, Jingjiang's south coastal area, Xinghua urban area, and Dainan Town; in contrast, for most of Xinghua's areas, Jiangyan's north area, Taixing's south part, and Jingjiang's northwest area, the soil erosion sensitivity is relatively low (Figure 3a).

RUSLE soil erosion model formula [34] is described by:

$$A = R \times K \times LS \times C \times P \tag{4}$$

wherein $A$ indicates soil erosion amount; $R$ indicates rainfall factor; $K$ indicates soil texture factor; $LS$ indicates length and slope factor; $C$ indicates vegetation coverage factor, and $P$ indicates water and soil conservation measure factor.

Agricultural measure is a factor closely related to human activities, and does not have much relation to the natural sensitivity of the ecosystem, so it is not taken as an evaluation index. According to Taizhou's actual situation and the importance degree of influence on soil erosion sensitivity, each factor is evaluated as follows: rainfall factor (0.2), soil texture (0.2), ground slope (0.2), and vegetation coverage (0.4). Each factor's internal evaluation index grading is executed by referring to the Ministry of Environmental Protection's standards and related documents [35–38] (Table 2).

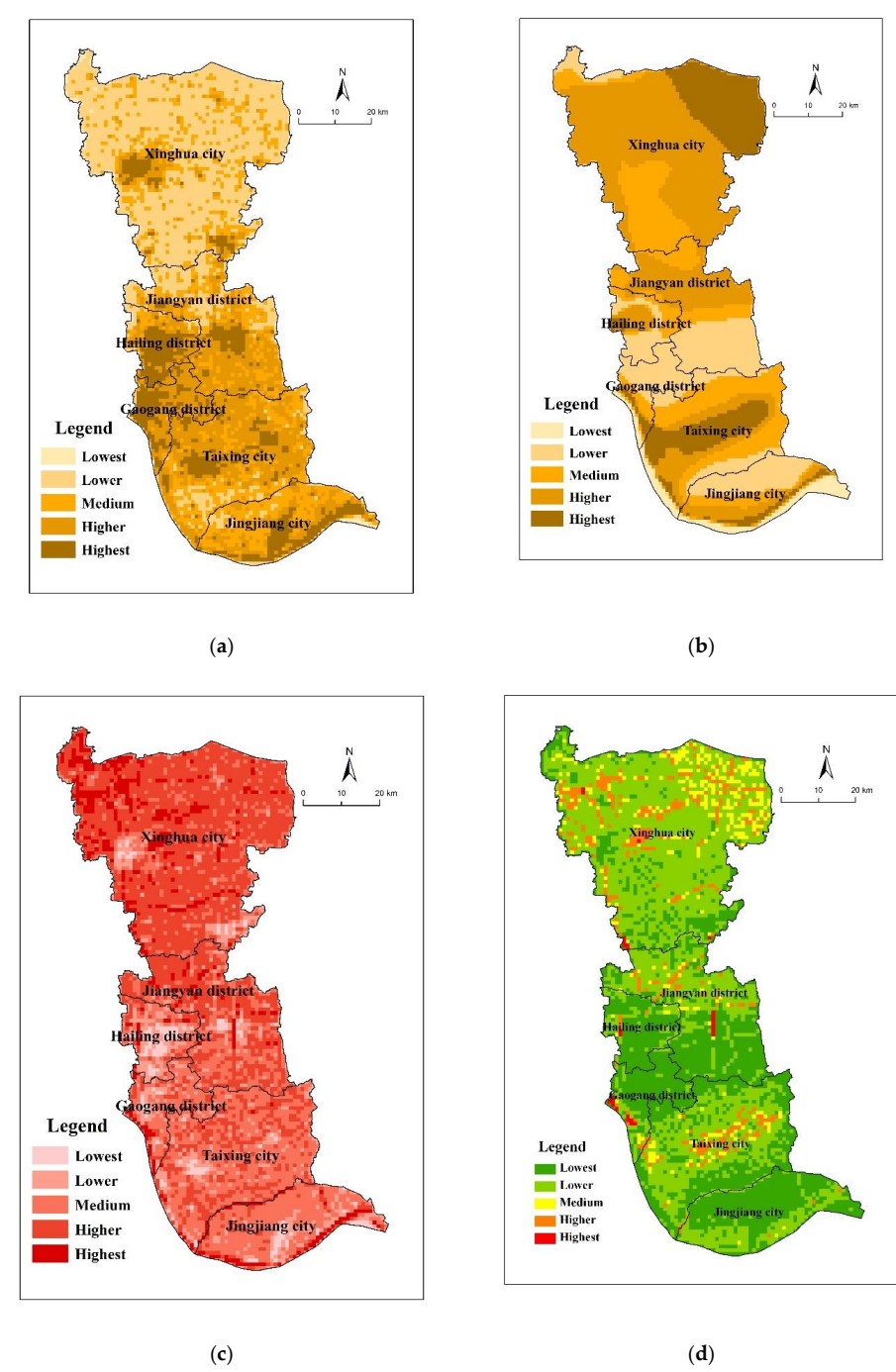

**Figure 3.** Ecological environment sensitivity evaluation. (**a**) Soil erosion sensitivity; (**b**) disaster vulnerability; (**c**) water environment sensitivity; (**d**) ecological environment sensitivity.

**Table 2.** Soil erosion sensitivity influential factors and grading.

| Sensitivity Factors | Not Sensitive | Lightly Sensitive | Moderately Sensitive | Hypersensitive | Extremely Sensitive |
|---|---|---|---|---|---|
| Rainfall erosion (mm) | <25 | 25–100 | 100–400 | 400–600 | >600 |
| Soil texture | Chad and sand | Coarse sandy soil, fine sandy soil, clay | Facing sandy soil, doras | Sandy loam soil, silt clay, and loam clay | Sandy silt soil, silt soil |
| Hypsography (°) | 0–5 | 5–10 | 10–20 | 20–45 | >45 |
| Vegetation coverage degree (%) | >90 | 70–90 | 50–70 | 30–50 | <30 |
| Grading value | 1 | 3 | 5 | 7 | 9 |

### 3.1.2. Vulnerability to Natural Disasters

Vulnerability is a multidimensional concept, which may be affected by the socio-economic conditions of the local communities. The vulnerability to natural disasters mainly reflects the possibility and rehabilitation difficulty of ecological and environmental problems in the development process, or the quantity of life or property losses possibly caused by improper development activities. Greater vulnerability to natural disasters indicates greater risk of disasters induced in this area, or the higher cost of ecological rehabilitation and environmental construction. Additionally, the availability of resources allocated to the post-disaster recovery would be relevant. For Taizhou, the disaster vulnerability was comprehensively evaluated through spatial superposition analyses of indices such as low-lying land selected, flood disaster, earthquake risk, ground setting, and collapse. On the whole, the areas of high natural disaster vulnerability, according to the index, are mainly distributed in the northeast part of Xinghua, the central part of Taixing, and the areas along the Yangtze River. Whereas, the areas where geological hazard occurs frequently are mainly at the northeast part of Xinghua, the central part of Taixing, and the area along the Yangtze River; the areas where flooding occurs frequently are mainly distributed in the Lixia River Basin; while in Jingjiang and Taixing's intersection area, Jiangyan's south area, and Gaogang's east area, the disaster vulnerability is relatively low (Figure 3b).

### 3.1.3. Water Environment Sensitivity

Water environment sensitivity indicates the regional ecosystem capacity to contain water pollutants in conditions of normal rainfall, namely the risk degree of water environmental pollution. Based on a projection of future trends in environmental protection and the current state of Taizhou's water environment pollution, the main concerns are to protect the urban drinking water source, guarantee water quality, and establish the water environment sensitivity evaluation index system (Table 3), giving priority to a drinking water protection area, water environment function zoning, and various land use types, and comprehensively evaluate the water environment pollution sensitivity index of each grid unit through the ArcGIS platform. Overall, water environment sensitivity is high in the northern part and low in the southern part of the whole city. The water environment sensitivity is relatively high in the areas along the Yangtze River, the No. 3 Water Works' drinking water source protection areas, water source protection area at the water transfer port of Yinjiang River, Zhonggan River Jiangyan drinking water source protection area, as well as the main rivers like the Yangtze River, Yinjiang River, and Jingtaijie River, etc. It is relatively low in the other areas. Xinghua has a relatively high risk of water environment pollution for reasons of high river and water system density, many lakes, and poor accessibility (Figure 3c).

**Table 3.** Quantified value of water environment sensitivity index.

| Index | Index and Weight | Factor Value |
|---|---|---|
| Water environment sensitivity | Drinking water source protection area (0.5) | First-grade protection zone 9 scores [1], 50 m buffer zone in protection area 7 scores, 100m buffer zone 5 scores |
| | Water quality objective of water environment function area (0.3) | Class-II water-quality rivers 7 scores, 50 m buffer zone 5 scores, and 100 m buffer zone 3 scores |
| | | Class-III water-quality rivers 5 scores, 50 m buffer zone 3 scores, and 100 m buffer zone 1 score |
| | | Class-IV and above water-quality rivers 3 scores, 50 m buffer zone 2 scores, and 100 m buffer zone 1 scores |
| | Land use type (0.2) | Water body 9 scores, paddy field 7 scores, dry farm 5 scores, forest land 3 scores, others 1 score |

[1] scores mean that different levels of conservation/water quality/land use have different levels of importance, and are assigned in descending order of importance.

### 3.1.4. Comprehensive Evaluation

Each evaluation unit's ecological environment sensitivity index is determined through comprehensive index evaluation, and is divided into high, relatively high, medium, relatively low, and low level by means of cluster analysis (Figure 3d). On the whole, the areas with a relatively high ecological environment sensitivity index are mainly distributed in the Xinghua's north areas, Jiangyan's northwest areas, and Taixing's central areas, as well as the main river areas like the Yangtze River, Yinjiang River, and Jingtaijie River, etc., while Hailing District, Gaogang District, Jiangyan's south areas, and Jingjiang's mid-north areas have relatively low ecological environment sensitivity.

### 3.2. Importance of Ecosystem Service Function and Evaluation

3.2.1. Importance of Water Source Conservation and Drinking Water Source Protection

Water source conservation areas have functions such as intercepting Yangtze River water, increasing soil infiltration, inhibiting evaporation, relieving surface runoff and increasing rainfall, etc., and play an important role in protecting the ecological safety of regional water [39–41]. Present water source conservation areas in Taizhou mainly include important wetlands, water channel maintenance areas, special ecological industry zones, and important fishery waters, etc. Drinking water source protection areas play a crucial role in guaranteeing clean drinking water, and include the drinking water source protection zones such as Taizhou No. 3 Water Works, Yinjiang River water transfer port, Zhongjiang River Jiangyan, Luting River, Yangtze River Pengqi Port, and Ganggu, etc. (Figure 4a).

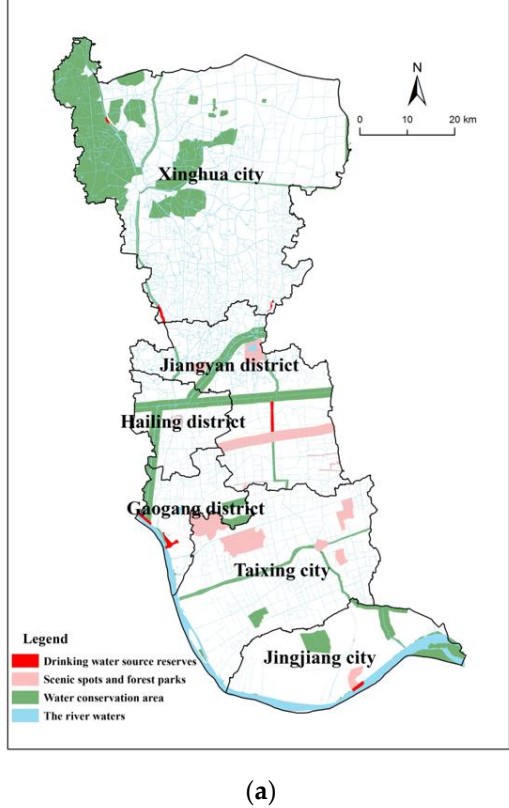

(**a**)

**Figure 4.** *Cont.*

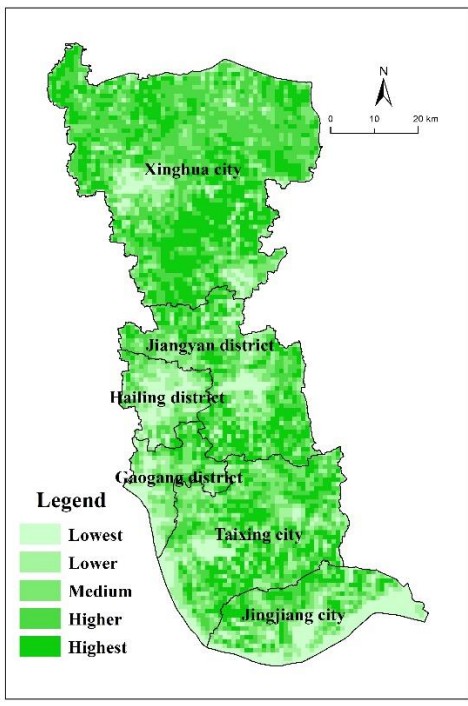

(**b**)

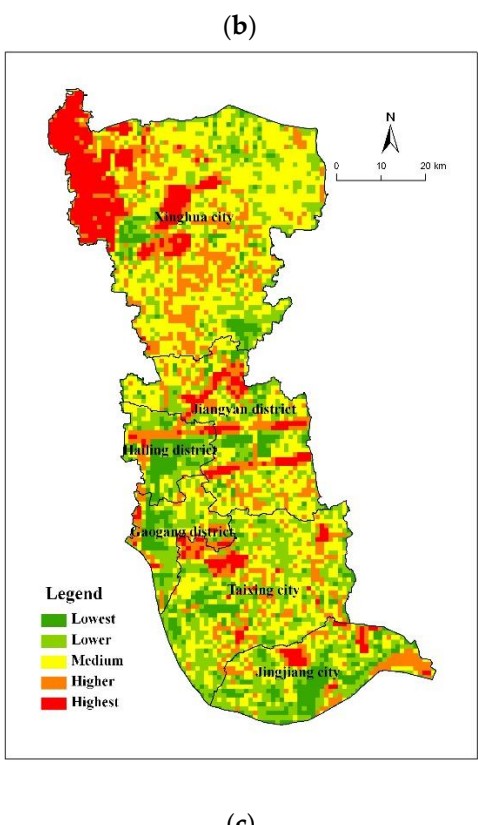

(**c**)

**Figure 4.** Ecosystem service function importance evaluation. (**a**) Important ecological function areas; (**b**) vegetation coverage index; (**c**) ecological function importance.

3.2.2. Importance of Natural and Cultural Heritage Protection

Natural and cultural heritage indicates the natural landscapes and cultural treasuries making contributions to human civilization and advancement in the area, and mainly includes scenic spots, forest parks, historical and cultural areas, cultural relics and historical

sites, etc., which provide human beings with service functions like adjustment, protection, buffer, absorption, reservation, education and culture, etc. Taizhou has 15 scenic spots and forest parks currently, covering a total area of around 194 km$^2$, including Huangchenghe Scenic Spot, Gushan Scenic Spot, Taizhou Yinjiang River Scenic Spot, Gixi Town Scenic Spot, Qinhu National Wetland Park, Zhoushan Forest Park, Baolaohu Forest Park, Taishan Park, and Taixing Park, etc. The historical and cultural heritage here mainly include Taixing Xinsijun Huangqiao Battle Memorial Hall, Jiangjiang Yuewang Temple—Liu Guojun's Former Residence, Taizhou CPC Zhejiang District Taixing Independent Branch Memorial Hall, and Zheng Banqiao Memorial Hall, etc. (Figure 4a).

### 3.2.3. Importance of Biodiversity Maintenance and Ecological Environment Protection

Biodiversity is beneficial for maintaining regional ecological balance, and providing good environmental conditions for human survival. According to the evaluation of the importance of biodiversity protection in different regions, the ecological regions belonging to the preferentially protected ecosystem and hot spots of ecological protection may be taken as areas having important functions on biodiversity maintenance and ecological environment protection. The vegetation coverage index reflects regional vegetation coverage and growth condition differences, and its index value increases along with the increase of biomass and could well evaluate regional biodiversity and ecological environment characteristics. In general, areas with high vegetation cover reflect the good condition of the local ecological environment, which can provide habitats suitable for living species to survive. It is relatively difficult to obtain biodiversity data, so this paper reflects biodiversity by dint of vegetation coverage. On the whole, Xinghua's vegetation coverage index is the highest. Whereas the vegetation coverage in the mid-north and central areas is high, in the areas along the Yangtze River, for urban construction land division, vegetation coverage is relatively dispersive (Figure 4b).

### 3.2.4. Comprehensive Evaluation

Each evaluation unit's ecosystem service function importance index is determined through comprehensive index evaluation, and is divided into high, relatively high, medium, relatively low, and low levels by means of cluster analysis (Figure 4c). On the whole, the areas with relatively high ecological importance are mainly distributed in most areas in the northwest part of Xinghua, the north and south areas of Jingjiang, the northwest area of Taixing, the west area of Gaogang, Hailing urban area, Jiangyan urban area, and Xinghua urban area and its southeast areas.

### 3.3. Socio-Economic Coercion Evaluation
#### 3.3.1. Pollution Discharge Pressure

Each town unit's industrial point-sourced pollution, poultry cultivation pollution, and urban domestic pollution are segmented into unit grids with ArcGIS software, and according to the proportion of their discharge, their weight is determined to be 0.4, 0.3, and 0.3, respectively, and the weighted sum is calculated based on standardized processing. The results show that the areas with a relatively high pollution pressure index are mainly distributed in Taizhou's areas along the Yangtze River. In Jingjiang urban area and south area along the river, Taixing's urban area and west area along the river, Gaogang's area along the river, Hailing urban area and its south area, and Jiangyan urban area, the pollution is relatively concentrated; Xinghua, except for Xinghua urban area, Dainan Town, Zhouzhuang Town, and Anfeng Town, etc., have a relatively high pollution index. Most other areas have a relatively low pollution index (Figure 5a).

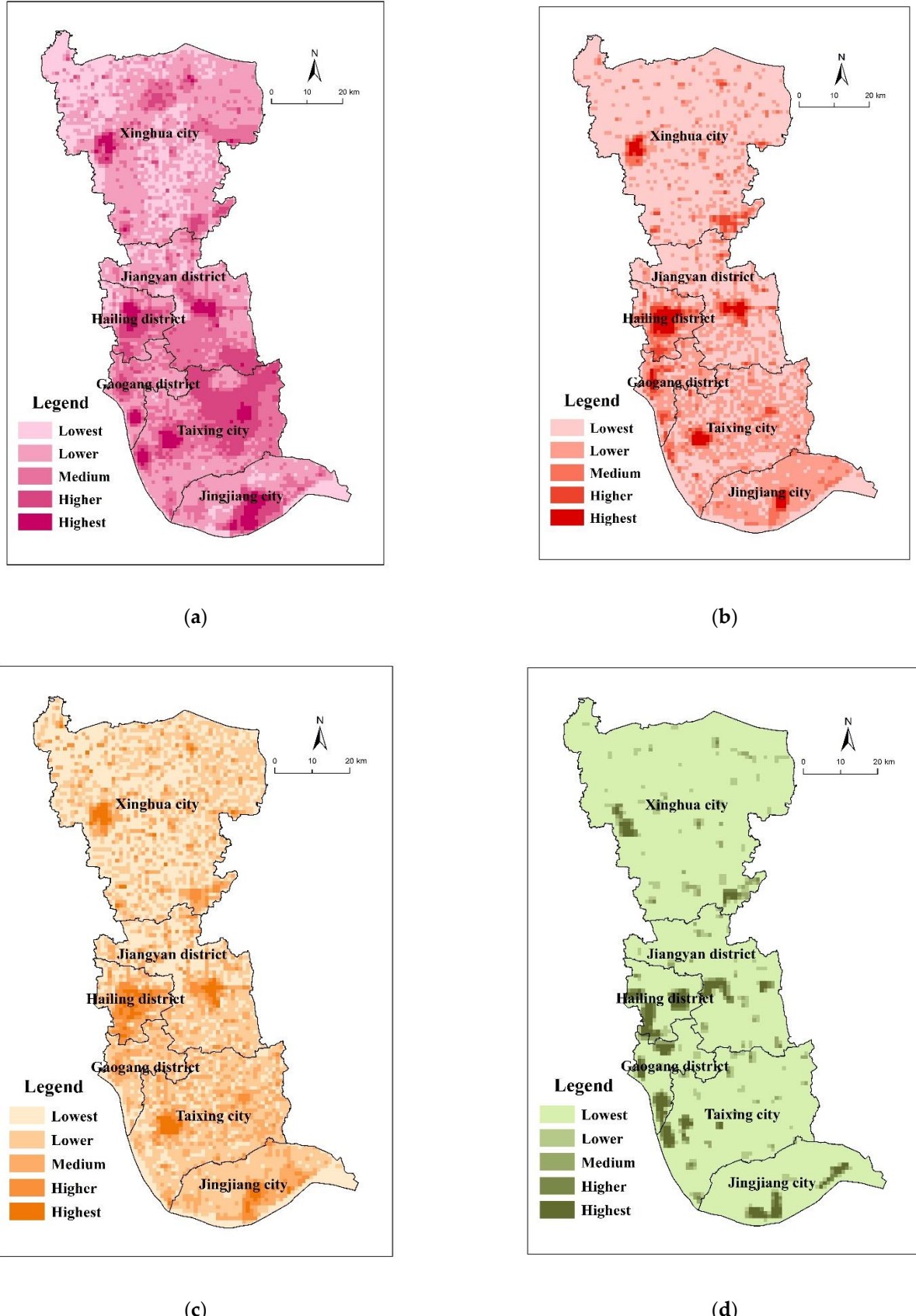

(**a**)

(**b**)

(**c**)

(**d**)

**Figure 5.** *Cont.*

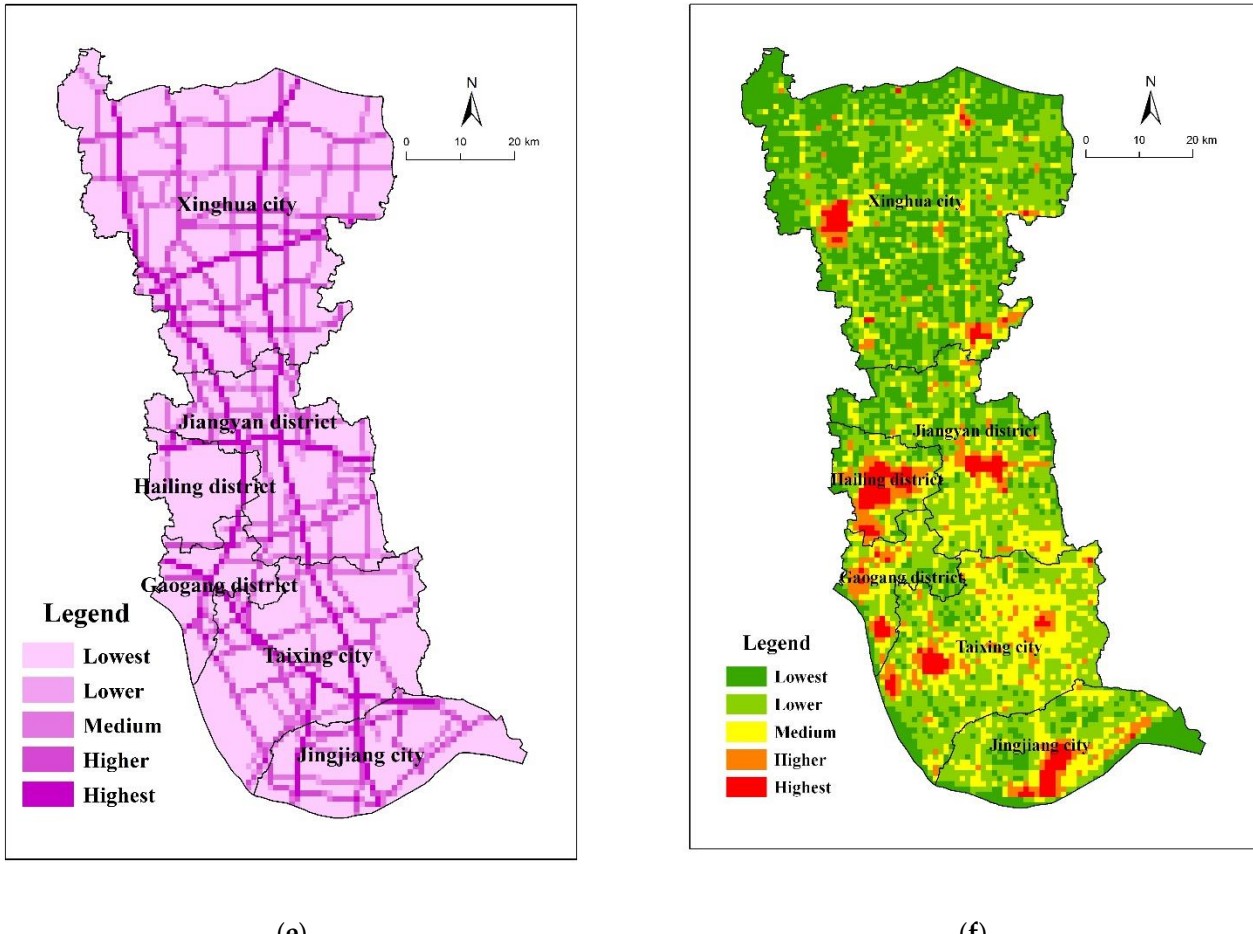

(**e**)                                    (**f**)

**Figure 5.** Ecological environment stress evaluation. (**a**) Pollution discharge pressure; (**b**) urban expansion pressure; (**c**) population gathering pressure; (**d**) industrial development pressure; (**e**) traffic layout pressure; (**f**) ecological environment stress.

### 3.3.2. Urban Expansion Pressure

Urban construction land expansion changes land use patterns, destroys original ecological structure, and lowers regional ecological environment capacity and certain ability to absorb pollutants [42]. Therefore, a higher proportion of unit area construction land produces greater pressure of ecological environment. Different types of construction land have differences in development strength. The construction land factors are segmented into unit grids with ArcGIS, and the value of ecological environment pressure brought by the construction lands in cities, towns, and rural areas is determined to be 3, 2, and 1, respectively. The results show that, in Hailing District, Gaogang District, Jiangyan urban area, Taixing urban area, Jingjiang urban area, Xinghua urban area, Dainan Town, Huangqiao Town, etc., the construction land proportion is relatively high, bringing relatively great pressure to the ecological environment; in most areas of Xinghua, and northwest area of Jiangyan, the construction land proportion is relatively low, having relatively small influence on ecological environment pressure (Figure 5b).

### 3.3.3. Population Gathering Pressure

The permanent population on unit grid land area is used to represent the degree of population gathering. Higher population gathering leads to greater resource consumption, more concentrated discharge of environment pollution, and greater pressure on the regional ecological environment [43]. The results show that in Hailing District, Jiangyan urban area, Taixing urban area, Jingjiang urban area, and Xinghua urban area, the degree of population

gathering is highest, bringing the greatest ecological environmental pressure to these areas; in contrast, in the northwest part of Xinghua and the northwest part of Jiangyan, the population gathering degree is the lowest, and so too is ecological environmental pressure (Figure 5c).

### 3.3.4. Industrial Development Pressure

Industrial parks will be the main focus for the gathering of industrial enterprises in Taizhou in the future. However, these sites will produce large quantities of pollutants such as waste water, waste gases, and solid wastes, and put pressure on the regional ecological environment [43]. According to the development grades, Taizhou's industrial parks are divided into key development blocks and general development blocks, and the value of ecological environment pressure caused by them is determined to be 2 and 1, respectively. Key development blocks include national, provincial, and municipal industrial development zones, while general development blocks are the industrial gathering areas developed by towns. The results show that the areas with a high industrial development pressure index are mainly distributed in the areas along the Yangtze River, Hailing District, Jiangyan urban area, Taixing urban area, Jingjiang urban area and Xinghua urban area, and include the High-Tech Pharmaceutical Park, Hailing Industry Park, and Jiangyan Economic Development Zone. The industrial gathering areas developed by each town have relatively high ecological environment pressure (Figure 5d); other areas have relatively low impact on ecological environment pressure.

### 3.3.5. Traffic Layout Pressure

Traffic facilities construction not only destroys the original ecosystem, but also produces traffic noise and motor vehicle tail-gas pollution. When analyzing potential ecological environment pressure, one should comprehensively consider the current state and trend of traffic facilities development, and recognize areas with environmental risk. According to the comprehensive traffic planning distribution chart in *Taizhou Urban Overall Planning (2010–2020)*, the impact of traffic facilities on ecological environment pressure is simulated by means of constructing a buffer belt with ArcGIS software. According to the differences of road grade, buffer zones with a space of 50m, 100m, and 200m are established at expressway, national highway, and provincial highway. By giving quantified value of the ecological pressure brought by different grades of roads, this paper analyzes the degree of traffic facilities' impact on the environment (Table 4). On the whole, at both sides of Beijing–Shanghai Expressway, Qidong–Yangzhou Expressway, Shanghai–Shanxi Expressway, Yancheng–Jingjiang Expressway, and the Shanghai–Zhenjiang–Yangzhou Expressway being planned, the impact on ecological environment is the highest; in the areas at both sides of national highways and provincial highways, the impact is the second highest; while in the other areas, the impact is the lowest (Figure 5e).

**Table 4.** Quantified value of traffic pressure factor.

| Road Type | Factor Value |
| --- | --- |
| Expressway | 50 m buffer zone 7 scores, 100 m buffer zone 5 scores, and 200 m buffer zone 3 scores |
| Arterial highway (national highway) | 50 m buffer zone 5 scores, 100 m buffer zone 3 scores, and 200 m buffer zone 1 score |
| Secondary road (provincial highway) | 50 m buffer zone 3 scores, 100 m buffer zone 2 scores, and 200 m buffer zone 1 score |

### 3.3.6. Comprehensive Evaluation

Each evaluation unit's ecological environment pressure index is determined through comprehensive index evaluation, and is divided into high, relatively high, medium, relatively low, and low levels by means of cluster analysis (Figure 5f). On the whole, the areas with relatively high ecological environment pressure are mainly distributed in Hailing District, Jiangyan urban area, Taixing urban area, Jingjiang urban area, Xinghua urban

area, Taizhou's riparian Gaogang urban area, Yonganzhou Town, Binjiang Town, Dongxing Town, and Xieqiao Town, as well as key towns such as Dainan Town and Huangqiao Town. The areas with relatively low ecological environment pressure are mainly distributed in the north and southwest parts of Xinghua, the northwest part of Jiangyan District, and at the intersection of Jingjiang and Taixing.

*3.4. Function Zoning*

Through matrix classification analysis of each grid evaluation unit's ecological environment sensitivity index, ecosystem service function importance index, and socio-economic coercion index, Taizhou is divided into four types of function zoning, namely ecological environment restoration zone, ecological economy bearing zone, ecological economy preservation zone, and ecological environment protection zone (Figure 6).

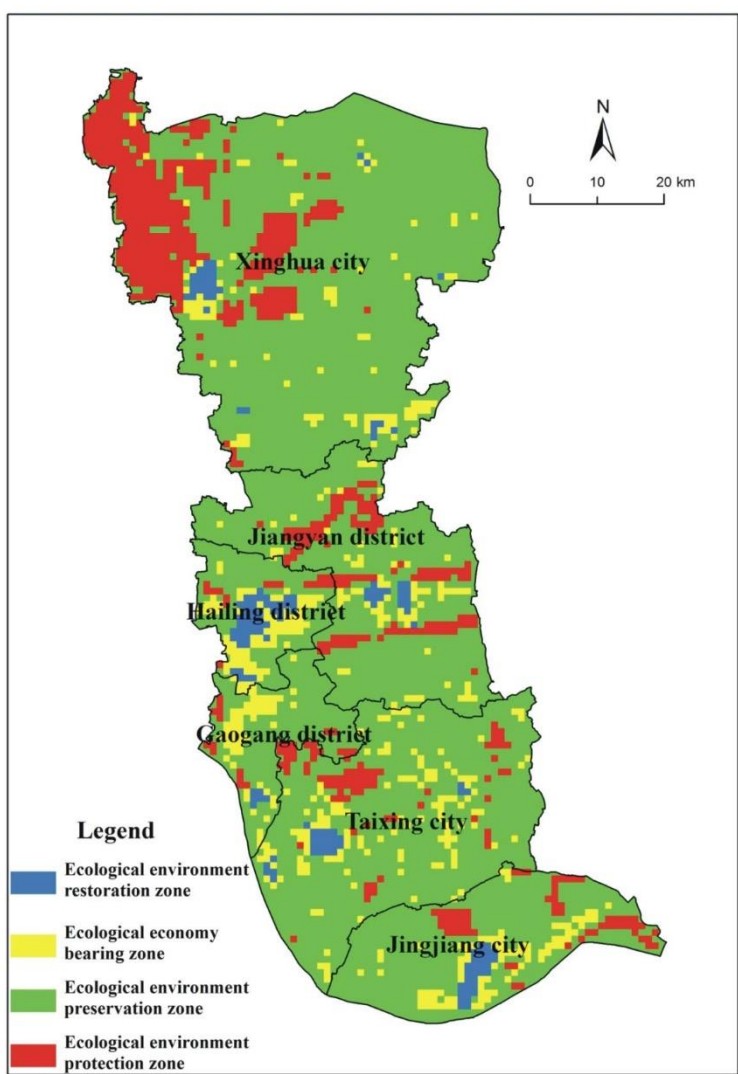

**Figure 6.** Four types of function zoning of ecological environment.

Ecological environment restoration zones indicate areas with low ecological environment sensitivity and small value of ecosystem service function importance, but heavy impact of human activities. These areas are mainly distributed in Hailing, Jiangyan, Taixing, Jingjiang and Xinghua urban areas, the central areas of Yong'an zhou Town and Binjiang Town along the river, and the core areas of the key towns like Dainan Town and Huangqiao Town, etc., covering an area of 162 km$^2$ and accounting for 2.79% of the total area. Such areas have a high concentration of human activities, a developed economy, and relatively

good environmental infrastructure, but great pressure of industrial pollution discharge and environmental risk prevention. Therefore, it is necessary to strengthen urban ecology construction and further restore and improve the ecological environment, expand the space of public facilities and green ecology, guide industrial clustered development in urban areas, implement more strict environmental access conditions and pollution discharge standards, remove heavy-pollution enterprises gradually, optimize industrial layout, raise the proportion of tertiary industry, and focus on developing modern services, high-tech, and tourism industries, in order to reduce gross pollutant discharge, and improve production and quality of life for the inhabitants.

Ecological economy bearing zones indicate areas with lower ecological environmental sensitivity, smaller value of ecosystem service function, and lower impact of human activities. Such areas are mostly distributed in Hailing Industry Park, High-Tech Pharmaceutical Park, Gaogang Technology Entrepreneurship Park, Binjiang Industry Park, Taixing Industry Park, China Fine Chemical Park Xinghua Economic Development Zone, and industry zones in townships and so on, covering an area of 528.54 km$^2$ and accounting for 9.11% of the total area. With much sufficient environmental capacity, more land for development, convenient traffic conditions, and better industrial foundation, such areas are the main region for clustered development of industrial economy in the future. Considering regional environment capacity, it is necessary to accelerate industrialization and urbanization, optimize urban spatial patterns, eliminate outdated industrial capacity, promote clean production, and implement industrial policy of replacing the old with the new, while preventing excessive land occupation, water consumption, and ecological environment pressure due to industrialization and urbanization.

Ecological environment preservation zones indicate the areas that are sensitive to ecological environment impact or have certain value of ecosystem service function, and are not subject to high stress of social economy pollution. This type of area is widely distributed over most of the range in Xinghua city and Jiangyan, Taixing, Jingjiang, Gaogang and Hailing District, covering an area of 4366.05 km$^2$ and accounting for 75.27% of the total area. Such areas give priority to semi-natural and artificial ecosystems, and mainly use the land in the form of agriculture and gardening production, with good grain production conditions. It is necessary to gradually eliminate the discharge of agricultural surface-sourced pollutants, control the application of fertilizers and pesticides in farmlands, strengthen the protection of soil and underground water environment, and ensure safe quality of soil environment and stable quality of food production. Moreover, developing urban and characteristic industries is a priority; while controlling scale under the capacity bearing limit of the ecological environment is a prerequisite.

Ecological environment protection zones indicate the areas that are extremely sensitive to ecological environment impact or have important value of ecosystem service function, and are subject to minor impact of human activities. Areas of this type mainly include domestic drinking water source protection areas, scenic spots, natural reserves, wetland protection zones, forest parks, and important clear water channel maintenance areas, etc., covering an area of 744.21 km$^2$ and accounting for 12.83% of the total area. Such areas have great value of ecological service function, relatively weak human society activities and ecological activities, and small stress of ecological safety, so they have an important status in maintaining Taizhou's ecological support system, and shall be specially protected. Priorities are to delimit key protection areas; strictly execute national and local rules and related standards; prohibit all development activities damaging the ecosystem; maintain the natural original state, and protect sensitive objects like rare species; and develop ecological relocation or ecological migration in areas unsuitable for habitation.

## 4. Discussion

Ecological environment function zoning emphasizes the degree of ecosystem service provision and stress, focusing more on the type, importance, and sensitivity of ecological functions [44]. Under the premise of natural resource conservation and sustainable development, such zones integrate and differentiate the sensitivity characteristics of different ecological functions to the impact of human activities, taking into account regional physical and geographical differentiation, biodiversity, and uneven development of human and nature [29]. Ecological environment function zoning is also the foundation and a prerequisite for implementing zoning management of the regional ecological environment. Nowadays, research on ecological function zoning of large-scale areas at the provincial level and above has become relatively mature and has important instructional significance for regional environmental protection [15]. These studies for large regions emphasize spatial heterogeneity arising from differences in environmental resources, ecological succession, and disturbances, mostly by identifying key factors, spatial patterns, and sensitivities of ecological processes to reveal regional differences in ecosystem functions, especially ecosystem supply capacity [29]. However, ecological environment functional zoning is designed to guide the sustainable development of regional economic–social–ecological complex systems, and therefore requires zoning that takes into account the impact of human activities and the demand for ecosystem services. Although some studies have been conducted on zoning based on ecological asset assessment as a way to consider human activities, they are still considered in terms of ecosystem supply [45]. Therefore, research on urban-level zoning based on ecological function areas needs to be further expanded and deepened. This specific research study aims to promote applied research and exploration of ecological environment function zoning at the municipal level. Furthermore, as the role of human beings in the ecosystem becomes gradually more prominent, the stress effects of human activities on the ecosystem should be considered [25]. Consequently, based on an analysis of the existing two ecological function zoning evaluation index systems [26,27], namely ecological sensitivity and ecosystem service function importance, this study properly adds an analysis of socio-economic coercion brought by human activities, considers an ecological environment function zoning index system at the municipal level in an explorative way, and divides municipal spaces into an ecological environment restoration zone, ecological economy bearing zone, ecological economy preservation zone, and ecological environment protection zone.

Considering the different characteristics for different types of areas between ecological environment, economic, and social bearing ability, in order to maintain the coordinated development of environmental quality and economy, it is necessary to take control measures distinctively. On the one hand, sensitive ecological environment areas must protect the native environment, expand effective agricultural utilization space to develop ecological agriculture, control agricultural surface-sourced pollution and maintain regional ecological balance, while it is necessary to further reduce the impact of human activities on the ecological environment, improve industrial access environment standard, execute the system of environmental impact assessment, and control development strength of human activities, implement ecological compensation policy, and provide direct ecological compensation to the residents making contributions to ecological protection. On the other hand, for economic supporting areas, a pressing issue of concern is to make full use of abundant capacity resources, accelerate industrialization and urbanization, cluster industrial economy and optimize urban spatial pattern, and also to protect existing environmental capacity, raise the utilization efficiency of regional resources and energy, control pollutants discharge, strengthen the construction of environmental infrastructure, and guarantee the stability and improvement of regional environmental quality, strengthen urban ecology construction, maintain the integration and coordination of artificial ecology with natural ecology, and raise regional ecological carrying capacity. This study is divided into four major zones: ecological environment restoration zone, ecological economy bearing zone, ecological environment preservation zone, and ecological environment protection zone.

For the four zones, different management strategies should be used to address different challenges to promote sustainable urban development (Table 5).

**Table 5.** Management strategies that can be adopted for different zones.

| Zones | Management Strategies |
|---|---|
| Ecological environment restoration zone | 1. Adopt targeted ecological restoration to further restore and improve the ecological environment.<br>2. Expand the area of green infrastructure and ecological space.<br>3. Guide the development of urban industrial clusters and optimize the industrial layout.<br>4. Implement stricter environmental access conditions and pollutant emission standards, gradually eliminate heavy polluting enterprises, and develop high-tech industries.<br>5. Increase the proportion of tertiary industries, focusing on the development of modern service industries. |
| Ecological economy bearing zone | 1. Appropriately accelerate the process of industrialization and urbanization on the basis of not exceeding the ecological and environmental carrying capacity.<br>2. Prevent over-occupation of land, consumption of water resources, and pressure on the ecological environment brought about by industrialization and urbanization.<br>3. Optimize the spatial pattern of the city, eliminate backward production capacity, promote clean production, and implement the industrial policy of replacing the old with the new. |
| Ecological environment preservation zone | 1. Control the application of chemical fertilizers and pesticides on farmland, and continuously reduce the emission of pollutants on the agricultural surface.<br>2. Strengthen soil and groundwater environmental protection to ensure the safe quality of the soil environment and stable quality of food production.<br>3. Focus on the development of urban specialty industries under the ecological environment carrying capacity. |
| Ecological environment protection zone | 1. Delineate key protected areas.<br>2. Maintain the natural pristine state and protect rare species and other sensitive objects.<br>3. Conduct ecological migration in uninhabitable areas. |

There are still shortcomings in this study. Although this study included anthropogenic impact indicators such as pollution discharge pressure, urban expansion pressure, population gathering pressure, industrial layout pressure, and traffic layout pressure, the evaluation indexes have certain limitations and cannot reflect the features of ecological environment function zoning in a more comprehensive and systematic way due to the lack of data. Further information on the ecological environment (especially biodiversity) must be collected to optimize and improve the existing index system of ecological environment function zoning. In addition, some studies have focused on the supply, demand, and flow of ecosystem services, and the mapping of ecosystem services has developed from supply capacity mapping to supply demand mapping [46,47], but this study is still relatively preliminary and there is a lack of more ecological environment function zoning oriented to the supply and demand of ecosystem services.

## 5. Conclusions

This research provides a new perspective for discussing ecological function zoning covering the entire city. It also provides theoretical evidence for the analytical framework of municipal ecological environment function zoning, enriching existing research. Meanwhile, with Taizhou as an example, and based on the approach of ecological environment function zoning, this study attempts to bring forward ecological environment protection measures and industrial development orientation for each functional area, and to provide a scientific basis for Taizhou's ecological city development and construction. Undoubtedly, the construction of a municipal ecological environment function zoning index system is a complicated subject, and its science and rationality need to be further verified in follow-up research. In addition, the evaluation unit is embodied in the form of a grid and is impossible to normally implement and operate in regional environmental management, so it is still necessary to further improve and integrate the evaluation unit with the administrative unit.

Finally, regarding differences in the ecological environmental characteristics of each region, the index system and application model put forward should be adjusted when it is used in other cities. In the study of specific urban cases, we need to treat different urban areas with discernment.

**Author Contributions:** Conceptualization, H.Z. and K.W.; methodology, X.J.; validation, H.Z; formal analysis, X.J. and B.G.; investigation, B.G.; data curation, H.Z.; writing—original draft preparation, X.J.; writing—review and editing, B.G.; visualization, X.J.; supervision, H.Z.; project administration, H.Z.; funding acquisition, H.Z. and K.W. All authors have read and agreed to the published version of the manuscript.

**Funding:** This research was funded by National Natural Science Foundation of China (71573250 and 41371178) and 135 Strategic Development Planning Project of Nanjing Institute of Geography and Limnology, Chinese Academy of Sciences (2012135006).

**Institutional Review Board Statement:** Not applicable.

**Informed Consent Statement:** Not applicable.

**Data Availability Statement:** The data presented in this study are available on request from the authors.

**Acknowledgments:** We would like to show our deepest gratitude to my friend, Patrick Steele, a respectable, responsible, and resourceful scholar at the Royal Town Planning Institute (RTPI), Manchester, UK, who has provided me with valuable guidance in every stage of the writing of this paper.

**Conflicts of Interest:** The authors declare no conflict of interest.

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
