# Peer review of "Evaluation and Functional Zoning of the Ecological Environment in Urban Space—A Case Study of Taizhou, China"

_sustainability, doi:10.3390/su14116619_

Round 1

Reviewer 1 Report

The research paper is really interesting and reflects the complex approach to the procedure of environmental zoning, but the choice of indicators may provoke several questions. The main question is whether is it possible the comprehensive assessment or no? May be it should be replaced by special assessment devoted to different environmental issue - air pollution, biodiversity conservation etc.

It is also unclear - how we can summarize the results of each of three modules of assessment? Is it possible to propose the integral zoning of urban area as a result of such work?

But as I have mentioned above such questions are typical for each comprehensive assessment and this paper propose us a new example of approach for urban area that can be useful for environmental research at regional scale.

The part Discussion may be added by comparison with results of similar research or methodological issues of complex assessment.

The title of Table 2 should be changed.

Author Response

Dear reviewer 1,

Thank you for your valuable comments on this article!

For the first question, we selected indicators mainly from three aspects: ecological environment sensitivity, ecosystem service function importance and ecological environment stress. We believe that the selection of indicators has covered the evaluation more comprehensively, for example, air pollution should be related to socio-economic factors such as urban industrial layout and road network density, and biodiversity is related to ecosystem function.

For the second question, we have explained how to summarize the evaluation of the three modules in 2.3 and 2.4, and the general zoning of the urban area into four functional zones in 3.4 (Figure 6).

Lastly, we have added to the discussion section and modified the title of Table 2.

Reviewer 2 Report

Comments for Manuscript 1709957

Abstract

The abstract is well-suited for the purpose. It was concisely written. Readers will be able to differentiate the four zones. When the terminological issues are cleared, the Abstract will be excellent for publication.

Line 11

Functional zoning seems an easy and common term. But readers from different contexts, i.e. cultures and geographical regions, can have their own interpretation. To establish a common understanding, readers may expect a definition of this key term in the main text. The authors would be expected to provide such definition. One or two sentences will be sufficient.

Line 13

This comment is also about the choice of term. In fact, terminology plays a vital role in aiding the understanding of the articles by readers. The "ecological environment" may require some attention. If urban areas are divided and planned based on their ecological functions, then just "ecology" would be a sufficient term. It is understood that ecology is composed of various environmental elements, e.g. soil, forest, etc. It seems that the authors are focusing on ecological functions, which may be more than the environment only. But if the authors would like to coin the term "ecological environment", it would still be fine if a working definition is provided. Just one or two sentences will be enough for communicating the definition.

Introduction

The Introduction section is based on a comprehensive review of the trends in urbanisation. Clear, concise research objective(s) should be stated in the ending paragraph.

Line 67 to 68

Based on the review, the authors can further elaborate what roles human beings play in the ecosystem in a Chinese research context. Maybe the authors can write one to two sentences about the possible roles played by human beings as examples.

Line 68 to 69

How about the research related to ecosystem in other parts of the world, e.g. Europe? It is suggested that the authors can add one to two sentences to describe the research progress in other parts of the world.

Line 78 to 81

Concerning zoning methods, the authors stated that there have been mature zoning methods. Perhaps the authors can cite a few studies on the methods of ecological zoning so that the readers can acquire basic knowledge before proceeding.

Line 84

Some Chinese cities or provinces were mentioned here. However, some readers may be located in cities outside China. Therefore, more background knowledge has to be provided. For example, the authors can name some polluting industries here. Also, the shift in industrial centres can be visualised using a map. Or maybe the authors can use local gross domestic products (GDP) figures to exemplify the spatial relocation of the industries. Perhaps the authors can follow one of the methods suggested above to enrich the description.

Line 82 to 89

The authors need to provide the objective(s) of the present research in the Introduction section. Readers shall read and remember the objective(s) as they go through the article. An appropriate objective acts as a solid backbone supporting the whole study. Just add a sentence to state explicitly the objective(s) of this paper.

Materials and Methods

Line 92 to 99

There is a lot of names of places in China, where the study area was located. However, not every reader comes from the city being studied. Thus, the authors should be certain that all mentioned names were featured in Figure 1, which is the map

Line 105 to 109

There are some numbers being cited here. It is believed that these numbers were obtained from the respective authorities. The authors should provide the references to the source of the information.

Line 147

Why is there a question mark in the list? Is it a typo? Please correct if this is a mistake

Line 164

Sufficient justifications must be provided for the determination of weightings. It may be tempting to provide a few references and mention that the weightings followed previous studies. If this is the case, the authors should evaluate the suitability of borrowing the weightings. But if the weightings were adjusted, please also state the rationale of doing so.

Results

Line 234

It would be better if the equation can be presented in the Methods section. Also, provide a reference for the equation

Line 238

Please refrain from using abbreviations such as this one

Line 240 to 244

Did the authors notice spatial variations in the physical geographical conditions which may lead to changes in the numbers mentioned in the latter half of the paragraph?

Line 248 to 262

Vulnerability is a multidimensional concept, which may be affected by the socio-economic conditions of the local communities. Also, the availability of resources allocated to the post-disaster recovery would be relevant. The authors may need to add one to two sentences to specify whether attention has been given to these aspects.

Line 283

The meaning of "scores" inside the table should be explained in the table caption

Line 327 to 328

It is understandable that a comprehensive biodiversity survey could not be carried out given the context of the study. But the authors must clarify the relationship between vegetation coverage and biodiversity. For instance, even if an area is entirely covered in tree monoculture, the biodiversity can be poor. Thus, the authors would be recommended to write one or two sentences to clarify such relationship

Line 434 to 490

The map was actually quite nice for illustrative purposes. However, it is believed that the process of producing the map involved some delicate calculations. As this is a quantitative research paper, the readers may expect to see some numbers in this important part of the paper. The complex principles of calculations can be omitted. Nonetheless, some relevant figures and numbers can be stated so that readers may find it easier to compare this study against other similar studies in the future in order to build a greater body of knowledge in this area.

Discussion

Line 500

Several stress factors were mentioned in this study. It is believed that the authors can add a list of factors here so that the results could echo with the previous research.

Line 508 to 526

Some management strategies related to the zones were stated in the paragraph. It would be better if the authors can create a table of the strategies corresponding to each zone. This can help readers who would like to quickly acquire ideas about environmental zoning and follow-up conservation actions.

Author Response

Dear reviewer 2,

Thank you for your specific and constructive comments and suggestions on this article, it has been very enlightening! Here are our revisions and explanations for each section.

Abstract: In this section, we have added the definition of functional zoning. In addition, since we believe that environmental pollution is also a major challenge for cities and should be considered in functional zoning, we still adopt many indicators of environmental pollution in the selection of indicators. So we believe that we cannot just replace ecological environment with ecology.

Introduction: The most important innovation of this paper is the consideration of the importance that human activities should have in the functional zoning of the ecosystem, and we think your comments are very relevant. Therefore, we added how human activities affect ecosystems in China. In addition, before reviewing the Chinese studies, we have actually reviewed the studies of the US and international organizations, and as suggested we have added the relevant literature, the European studies on ecosystem zoning and the methods of ecological zoning. In the last paragraph, we add data on above-designated enterprises in the mentioned regions and highlight the theoretical contribution to the research objectives of this paper.

Materials and Methods: In this section, we have added or corrected the suggestions you made. In addition, in Figure 1, we have shown the location map of Taizhou, where Yangtze River Delta is in the lower left small map, the relationship between Taizhou and other surrounding cities (Yangzhou, Zhenjiang, Wuxi, Suzhou, Nantong and Yancheng) has been shown in the right map, and the administrative divisions within Taizhou have been shown in the right map. Regarding the determination of index weights in the article mainly considering the variability and importance, in order to make the weight setting more accurate and scientific, the hierarchical analysis method (AHP) and Telfer determination method are used. The AHP combines quantitative analysis with qualitative analysis, uses the experience of decision makers to judge the relative importance between the criteria of whether each measurement goal can be achieved among them, and gives the weights of each criterion for each decision option in a reasonable way, and uses the weights to find out the order of merit of each option. Among them, some experts in the field of ecological environmental protection were invited to score the importance of their indicators, and 80 questionnaires were distributed for investigation, and finally the corresponding weight values of the indicators were obtained by comprehensive evaluation.

Results: The issues and suggestions identified in this section have been modified and supplemented accordingly, clarifying the meaning of scores in Table 3 and providing additional explanations on the relationship between biodiversity and vegetation cover. In addition, since the topography of Taizhou is mainly plain (as explained in 2.1), we believe that the physical geography can be regarded as basically the same. And the calculation of 3.4 is actually realized through the basis of the calculation of 3.1-3.3, which has been explained in the first paragraph of 3.4, and the calculation and zoning methods are explained in 2.3-2.4. The results of the calculations are shown through Figure 6, which has also been illustrated in the beginning part of the second to fifth paragraphs, how much area of each type of zoning and what percentage of the total land.

Discussion: While previous ecological sensitivity and ecosystem service function importance have been used for zoning, the biggest innovation of this paper is to consider that human activities at the urban level can cause some stress on ecological protection. Some of the previous studies have been cited and are available for the reader's reference. In addition, Table 5 is added to reflect management strategies that can be adopted for different zones.

Reviewer 3 Report

This research article is important for ecological environment management and protection. There will be a large audience interested in the topic. The methodology, data processing, and results presentation are very clear. Please see some personal comments:

  1. All figures should be improved, please use high-resolution figures and pictures.

Line 14, ‘the paper’ please use ‘this study’.

Line 140-142, please add references.

Line 147, ‘COD? discharge’,  what is ‘COD?’.

Line 151, ‘analytic hierarchy process (AHP) and the Delphi Way’, please explain this. How do you choose the ten professors and relevant experts?  Additionally, how many questionnaires did you take? This place needs more details and explanation.

Line 162, Need a reference or more explanation for ‘prepare a weight judgment matrix table’

Line 192-201, The authors provide a very detailed data source, but there still are some details that need to be added. For example, ‘population and economic data’ from the city statistical yearbook, but did not give the information of year, data resolution and types of data (eg, GDP for economic data?). In line 195, authors mentioned remote sensing image and land use map but did not talk about the resolution of the image and how many types of land use in your data? Did you classify the images yourself? Or just borrow on others’ work? Please add some explanation.

Line 209, In Kernel module in ArcGIS, there are many factors affecting the kernel analysis results including search radius, analysis method, and population field. Please give more details about this part.

Line 217, ‘ArcGIS software’, please add reference. Which version do you use?

Line 218, What is the wide range of different buffer levels and why do you choose them?

Line 245, please add a title for Table 2.

Author Response

Dear reviewer 3,

Thank you for your questions about the research methodology of this paper! The following is a description of the revisions to this paper.

Line 14, ‘the paper’ please use ‘this study’.

Line 140-142, please add references.

Line 147, ‘COD? discharge’, what is ‘COD?’.

Line 245, please add a title for Table 2.

  1. The comments of L14, L140-142, L147, and L245: we corrected the writing problems in the paper and added literature references.

Line 151, ‘analytic hierarchy process (AHP) and the Delphi Way’, please explain this. How do you choose the ten professors and relevant experts?  Additionally, how many questionnaires did you take? This place needs more details and explanation.

Line 162, Need a reference or more explanation for ‘prepare a weight judgment matrix table’

  1. The comments of L151 and L162: we supplemented the index weight determination method, which was supplemented by references and description of the implementation method.

Line 192-201, The authors provide a very detailed data source, but there still are some details that need to be added. For example, ‘population and economic data’ from the city statistical yearbook, but did not give the information of year, data resolution and types of data (eg, GDP for economic data?). In line 195, authors mentioned remote sensing image and land use map but did not talk about the resolution of the image and how many types of land use in your data? Did you classify the images yourself? Or just borrow on others’ work? Please add some explanation.

Line 209, In Kernel module in ArcGIS, there are many factors affecting the kernel analysis results including search radius, analysis method, and population field. Please give more details about this part.

Line 217, ‘ArcGIS software’, please add reference. Which version do you use?

Line 218, What is the wide range of different buffer levels and why do you choose them?

  1. The comments of L192-201, L209, L217, and L218: we revised the data sources and processing section, mainly for specific details when it comes to data sources and data processing, such as demographic and economic data, remote sensing images, use of Kernel module, ArcGIS version, and so on.